# Image-Rejected Multi-Band Frequency Down-Conversion Based on Photonic Sampling

**Liuzhu Xu** [1,2], **Di Peng** [1,2,*], **Yuwen Qin** [1,2,3,*], **Jianping Li** [1,2], **Meng Xiang** [1,2], **Ou Xu** [1,2] and **Songnian Fu** [1,2]

1   Institute of Advanced Photonics Technology, School of Information Engineering, Guangdong University of Technology, Guangzhou 510006, China
2   Guangdong Provincial Key Laboratory of Information Photonics Technology, Guangdong University of Technology, Guangzhou 510006, China
3   Synergy Innovation Institute of GDUT, Heyuan 517000, China
*   Correspondence: dipeng@gdut.edu.cn (D.P.); qinyw@gdut.edu.cn (Y.Q.)

**Abstract:** An image-rejected multi-band frequency down-conversion scheme is proposed and experimentally demonstrated based on photonic sampling. The multi-band radio-frequency (RF) signals to be processed are copied into two replicas in quadrature, which are then sampled by an ultra-short optical pulse train via a polarization-multiplexed modulator. After polarization demultiplexing and detection using a pair of low-speed photodetectors, the multi-band RF signals are simultaneously down-converted to the intermediate frequency (IF) band. The image components can be suppressed by quadrature coupling the two generated IF signals via an electrical 90° hybrid coupler (HC). In the experiment, multi-band RF signals in the frequency range of 6 GHz to 39 GHz are down-converted to the IF band below 4 GHz using a local oscillator (LO) signal at 8 GHz to generate the ultra-short optical pulse train. Image rejection is achieved in the digital domain using digital signal processing to compensate for the amplitude and phase mismatch between the two IF signals and to implement quadrature coupling. In addition, through using an electrical phase shifter, an electrical attenuator, and an electrical 90° HC to achieve quadrature coupling of the two IF signals, image-rejected multi-band frequency down-conversion is also verified in the analog domain.

**Keywords:** frequency down-conversion; image rejection; photonic sampling; cavity-less optical pulse source

## 1. Introduction

Frequency down-converters are critical components in wireless communication and radar systems that transfer the received radio frequency (RF) signals to the intermediate frequency (IF) band to facilitate data acquisition and processing [1]. To satisfy the demands of the fast mobile traffic growth and the multi-function target detection with an anti-jamming effect, multiple RF bands are simultaneously enabled in such applications [2,3]. Therefore, multi-band frequency down-conversion is urgently required to accommodate to the RF signals in different frequency bands. Additionally, the image interference will be introduced in the frequency down-conversion procedure, since both the RF components and the corresponding image components are symmetrical with respect to the frequency of the local oscillator (LO) signal, which are down-converted to an identical IF band. Although the image interference can be avoided by using an electrical bandpass filter to select the target RF signal before down-conversion, it is challenging to achieve a wide-range tunable RF filter with a steep roll-off factor, constraining the operation bandwidth and the flexibility of the system. In fact, electrical mixers based on either Hartley or Weaver architecture are effective for realizing frequency down-conversion with image rejection [4,5]. Therefore, the crucial point of the Hartley method is to obtain a pair of IF signals with an orthometric characteristic, which can be realized by using two parallel electrical mixers to mix either a pair of receiving replicas in quadrature with the LO signal, or a pair of LO signals in

quadrature with the received signal. However, due to the limited octave and the high nonlinearity in the electronic devices, the operation bandwidth of a single electrical mixer with image rejection is not in a position to cover multiple frequency bands.

Microwave photonic technology is recognized as a promising candidate to circumvent the electronic bottleneck resulting from its large operation bandwidth. Most of the broadband photonic-assisted frequency down-conversion schemes are realized based on electro-optic modulation, where the IF signal is acquired through mutual beating between the specific modulation sidebands corresponding to the RF signal and the LO signal at a photodetector. Under such a construction, the orthometric characteristic for eliminating the image interference can be introduced by employing either the optoelectronic method or the all-optical method [6–31]. In the optoelectronic-based scheme, a pair of RF/LO signals in quadrature are firstly produced by using an electrical 90° hybrid coupler (HC). Then, a pair of orthometric IF signals are obtained after photodetection. Thus, the image components can be suppressed after the quadrature coupling of the two orthometric IF signals from the parallel photonic links [6–8]. As a comparison, the 90° phase difference between the parallel links can also be produced in an all-optical way, such as by adjusting the direct-current (DC) voltage of the electro-optic modulator [9–12], tuning the polarization controller (PC) after polarization division multiplexing [13–21], and adopting a 90° optical HC [22–29]. All of these frequency mixers realize excellent image rejection results. Nevertheless, in order to remove the undesirable modulation sidebands, optical spectrum selectors, such as an optical filter, a wavelength division multiplexer, and an asymmetric Mach–Zehnder interferometer, are essential for decreasing the operation bandwidth and the flexibility of the all-optical-based scheme [9–19,25–29]. The filter-free mixer with image rejection has already been proposed using a dual-parallel Mach–Zehnder modulator (DPMZM), with the assistance of an electrical 90° HC, for the purpose of suppressing the undesired sidebands via carrier-suppressed single-sideband (CS-SSB) modulation [20–24]. In [30], a compact scheme using a multi-functional mixer with image rejection, spur suppression, and dispersion immunity has been demonstrated, without using any optical spectrum selectors. Although existing image-rejection mixers based on microwave photonic technology are capable of realizing the image rejection for an RF signal in a large frequency range, the wideband characteristic of the down-conversion relies on a wide-range tunable LO signal generator, which inevitably increases the complexity of the receiver.

In this paper, a multi-band frequency down-conversion scheme is proposed and experimentally demonstrated based on photonic sampling. This scheme is featured with image rejection, and it does not require any electrical or optical filter. Benefiting from photonic sampling via an ultra-short optical pulse train, only a low-frequency LO signal is needed to achieve multi-band frequency down-conversion in a large frequency range. In the experiment, multi-band RF signals in the frequency range of 6 GHz to 39 GHz are simultaneously down-converted to the IF band below 4 GHz by using an LO signal at 8 GHz. Through using digital signal processing to compensate for the amplitude and phase mismatch between two parallel branches, the image components in the whole operation bandwidth are suppressed to be below the noise floor. In addition, the capacity of real-time image rejection is also verified in the analog domain.

## 2. Operation Principle

Figure 1 shows the schematic diagram of the proposed filter-free image-rejected multi-band frequency down-conversion based on photonic sampling. To facilitate understanding the signal conversion process in the optical and electrical domains, Figure 1a–f shows the signal waveforms and spectra at the corresponding nodes of the system. The ultra-short optical pulse train used for photonic sampling is generated by a cavity-less optical pulse source. The optical source is composed of a microwave frequency synthesizer, a power divider, an electrical amplifier (EA), an electrical phase shifter (EPS), a distributed feedback laser diode (DFB-LD), an electro-optic Mach–Zehnder modulator (MZM), an electro-optic phase modulator (PM), and a spool of single-mode fiber (SMF), as shown in Figure 1. The

operation principle of the cavity-less optical pulse source is briefly introduced as follows. The LO signal at $f_{LO}$ from the microwave frequency synthesizer is divided into two branches. One part of the LO signal is modulated onto the continuous-wave (CW) light at $f_0$ from the DFB-LD via the MZM biased at its linear transmission point to produce an initial optical pulse train with a repetition rate of $f_{LO}$, where the peak-to-peak voltage of the LO signal is set to be almost equal to the half-wave voltage of the MZM. The other part of the LO signal is amplified and used to drive the PM, where its phase is properly adjusted through the EPS until the temporal profile of the LO signal aligns with the optical pulse in the time domain. Under this condition, an approximate quadratic phase is introduced into each optical pulse. After propagation through a spool of SMF with a proper group velocity dispersion (GVD) value, the chirped optical pulse train is compressed into an ultra-short optical pulse train with a repetition rate of $f_{LO}$, whose spectrum is an optical comb centered at $f_0$, and with a frequency interval of $f_{LO}$, as shown in Figure 1a. As for the frequency down-conversion system, the generated ultra-short optical pulse train enters a dual-polarization dual-drive Mach–Zehnder modulator (DP-DDMZM), where the RF signal to be down-converted is optically sampled via linear electro-optic modulation. Thus, the RF signal at $f_{RF}$ and the possible image signal at $f_{IM}$ (supposing $f_{RF} > f_{IM}$) are divided into two parts with identical amplitudes, but with a 90° phase difference, by using an electrical 90° HC. Then, each part of the signal is copied into two replicas with a reversed phase by using an electrical 180° HC and is applied to DDMZM1 and DDMZM2, respectively. Both DDMZM1 and DDMZM2 work at the quadrature point to achieve linear electro-optic modulation. From the perspective of spectrum, the signal to be down-converted is modulated onto each tooth of the optical frequency comb at $f_0 + kf_{LO}$ ($k = 0, \pm1, \pm2 \dots$ ), and the modulated optical spectra in two polarizations are presented in Figure 1b,c. After passing through a 90° polarization rotator (PR), the sampled optical pulse trains from the two DDMZMs are polarization multiplexed using a polarization beam combiner (PBC). Then, the sampled optical pulse trains are polarization demultiplexed via a polarization beam splitter (PBS) and are detected by using two photodetectors (PDs), respectively. The RF signal and the possible image signal are down-converted to the IF band, and the electrical spectra are shown in Figure 1d,e. Here, $f_{IF\_RF}$ and $f_{IF\_IM}$ denote the IF signals down-converted from the RF signal and the image signal, respectively. The phase difference of the IF signals down-converted from the RF signal between the two channels is −90°, while it is 90° for the IF signals down-converted from the possible image signal. Through combining the two groups of the IF signals via an electrical 90° HC, the IF signals down-converted from the possible image signal cancel each other, while the IF signals down-converted from the RF signal superpose in phase, as shown in Figure 1f. Hence, filter-free image-rejected frequency down-conversion is achieved. In addition, due to the use of photonic sampling, RF signals in a large frequency range can be simultaneously down-converted to the Nyquist bandwidth below $f_{LO}/2$. Therefore, the proposed scheme can realize filter-free image-rejected frequency down-conversion, in which only a low-frequency LO signal is required.

Mathematically, the MZM and the PM in the optical pulse source are driven by single-tone microwave signals of $V_{LO1}(t) = V_{LO1} \cdot \cos(\omega_{LO}t)$ and $V_{LO2}(t) = V_{LO2} \cdot \cos(\omega_{LO}t)$, respectively. Here, $V_{LO1}$ and $V_{LO2}$ are the voltage magnitudes of the microwave signals applied to the MZM and the PM, respectively, and $\omega_{LO}$ is the angular frequency of the microwave signal. Through properly tuning the EPS, the signal profile applied to the PM aligns with each optical pulse. After propagation through a spool of SMF, the optical field of the ultra-short optical pulse train output from the cavity-less optical pulse source can be written as [31]

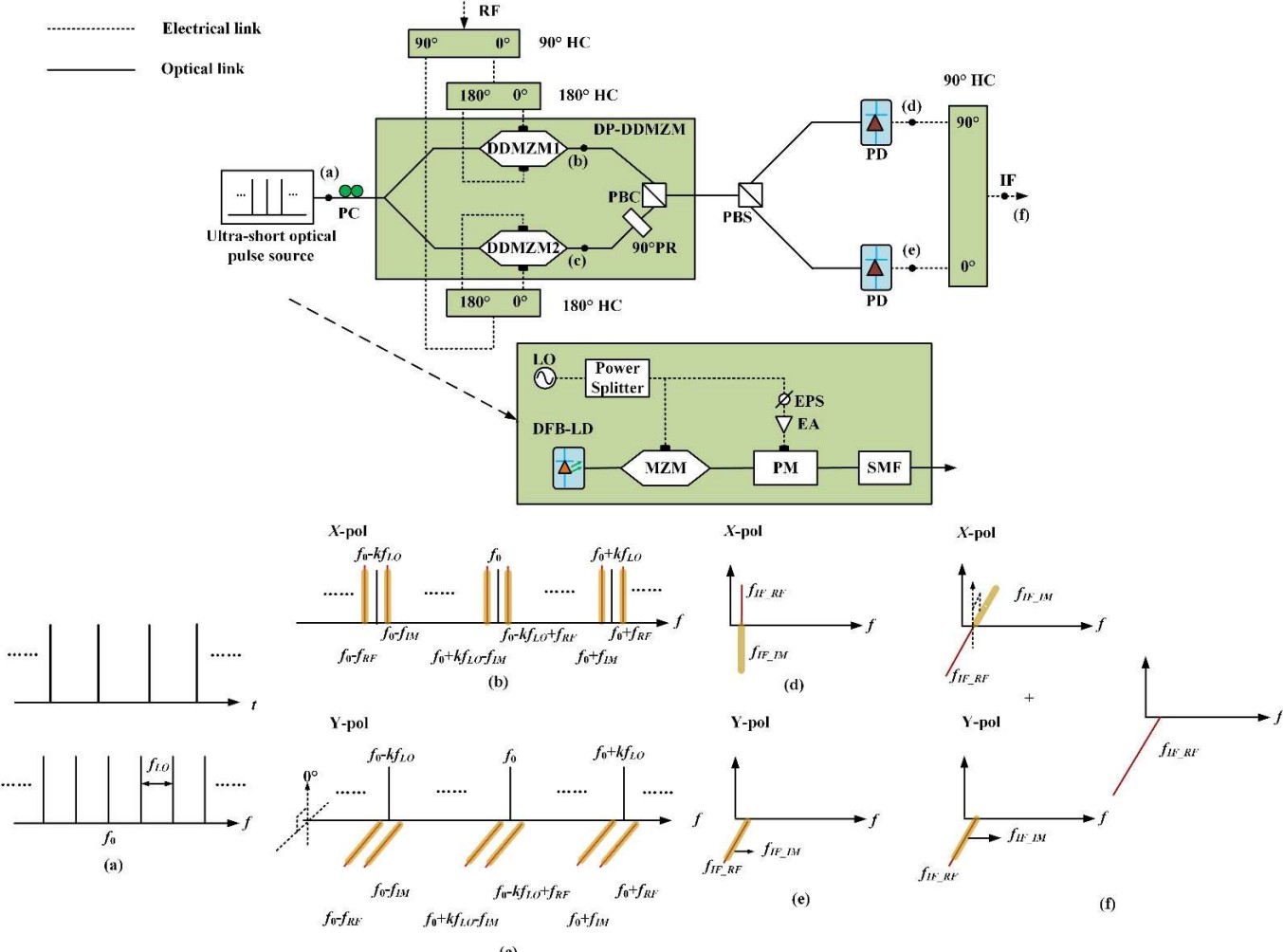

**Figure 1.** Schematic diagram of the proposed image-rejected multi-band frequency down-conversion. (**a**) Temporal waveform and optical spectrum of the sampling optical pulse train from the cavity-less optical pulse source; (**b**) optical spectrum from DDMZM1; (**c**) optical spectrum from DDMZM2; (**d**) electrical spectrum after photodetection in the upper branch; (**e**) electrical spectrum after photodetection in the lower branch; (**f**) electrical spectrum after quadrature coupling. PC: polarization controller; RF: radio-frequency; DP-DDMZM: dual-polarization dual-drive Mach–Zehnder modulator; 90° PR: 90° polarization rotator; PBC: polarization beam combiner; PBS: polarization beam splitter; PD: photodetector; HC: hybrid coupler; DFB-LD: distributed-feedback laser diode; MZM: Mach–Zehnder modulator; PM: phase modulator; SMF: single-mode fiber; LO: local oscillator; EPS: electrical phase shifter; EA: electrical amplifier.

$$E_{\text{SMF}}(t) = E_0 \sum_{n=-\infty}^{\infty} \sum_{k=-\infty}^{\infty} \cos\left(\frac{\varphi_1}{2} + \frac{n}{2}\pi\right) j^{(k-n)} J_n\left(\frac{m_1}{2}\right) J_{k-n}(m_2) e^{j(k-n)\pi} e^{j(k\omega_{LO})t} e^{j\frac{\beta_2}{2}Lk^2\omega_{LO}^2} \tag{1}$$

where $E_0 = E \cdot e^{jw_0 t}$ is the optical filed of the CW light from the DFB-LD. $m_1 = \pi V_{LO1}/V_{\pi 1}$ and $m_2 = \pi V_{LO2}/V_{\pi 2}$ are the modulation indices of the MZM and the PM, respectively, where $V_{\pi 1}$ and $V_{\pi 2}$ are the half-wave voltages of the MZM and the PM, respectively. $\varphi_1 = \pi V_{biasup}/V_{\pi 1\_DC}$ is the bias phase shift of the MZM, where $V_{bias1}$ and $V_{\pi 1\_DC}$ are the bias voltage and the DC half-wave voltage of the MZM, respectively. $J_n(x)$ is the $n$th-order Bessel function of the first kind. $L$ and $\beta_2$ are the length and the GVD coefficient of the SMF, respectively. The signal to be processed, i.e., $V_{sig}(t) = V_{RF} \cdot \cos(\omega_{RF} t) + V_{IM} \cdot \cos(\omega_{IM} t)$, is applied to the DP-DDMZM and optically sampled by the ultra-short optical pulse train (assuming $\omega_{RF} > \omega_{IM}$), where DDMZM1 and DDMZM2 are biased at their quadrature

points. $\omega_{RF}$ and $V_{RF}$ are the angular frequency and the voltage magnitude of the RF signal, respectively. $\omega_{IM}$ and $V_{IM}$ are the angular frequency and the voltage magnitude of the possible image signal, respectively. The output optical field of the DP-DDMZM in the time domain can be expressed as

$$E_{DP-DDMZM}(t) = \begin{bmatrix} E_x \\ E_y \end{bmatrix} = E_{SMF}(t) \begin{bmatrix} \cos\left(\frac{m_3}{2}\sin(\omega_{RF}t) + \frac{m_4}{2}\sin(\omega_{IM}t) + \frac{\pi}{4}\right) \\ \cos\left(\frac{m_3}{2}\cos(\omega_{RF}t) + \frac{m_4}{2}\cos(\omega_{IM}t) + \frac{\pi}{4}\right) \end{bmatrix} \quad (2)$$

where $m_3 = \pi V_{RF}/V_\pi$ and $m_4 = \pi V_{IM}/V_\pi$ are the modulation indices corresponding to the RF signal and the image signal in the DP-DDMZM, respectively. $V_\pi$ is the half-wave voltage of the DP-DDMZM. Under small-signal modulation, the output optical field of the DP-DDMZM can be simplified as

$$E_{DP-DDMZM}(t) \approx E_{SMF}(t) \begin{bmatrix} a + b\sin(\omega_{RF}t) + c\sin(\omega_{IM}t) \\ a + b\cos(\omega_{RF}t) + c\cos(\omega_{IM}t) \end{bmatrix} \quad (3)$$

where $a$, $b$, and $c$ are the amplitude coefficients of the optical carrier, the 1st-order modulation signal corresponding to the RF signal, and that corresponding to the image signal, respectively. After polarization demultiplexing and photodetection, the output currents from the two photodetectors can be calculated as

$$\begin{bmatrix} i_x(t) \\ i_y(t) \end{bmatrix} \propto \begin{cases} \sum_{N=1}^{\infty} A_N\cos(N\omega_{LO}t) + Q_1 A_N\sin((\omega_{RF} - N\omega_{LO})t) - Q_2 A_N\sin((N\omega_{LO} - \omega_{IM})t) \\ \sum_{N=1}^{\infty} A_N\cos(N\omega_{LO}t) + Q_1 A_N\cos((\omega_{RF} - N\omega_{LO})t) + Q_2 A_N\cos((N\omega_{LO} - \omega_{IM})t) \end{cases} \quad (4)$$

where $A_N$ is the amplitude of the frequency-multiplied components at $N\omega_{LO}$. $Q_1$ and $Q_2$ are the amplitude coefficients of the IF components generated from the beating between the RF/image signal modulation sidebands and the corresponding frequency teeth of the optical pulse source, respectively. Equation (4) indicates that the RF signals and the corresponding image signals in the two branches are simultaneously down-converted to the IF band below $f_{LO}/2$ when $N = \lfloor(\omega_{RF}/\omega_{LO}) + 0.5\rfloor$. Thus, $\lfloor x \rfloor$ represents the rounding down of $x$. Although the IF signals down-converted from the RF and the image signals at $f_{IF\_RF}$ and $f_{IF\_IM}$ are aliased in each branch, with an identical frequency and amplitude, their phases are $\pm 90°$, respectively. Therefore, after combining the two output optical currents via an electrical 90° HC to introduce a 90° phase shift in the first branch, the output current of the 90° HC is calculated as

$$i(t) = i_x(t) \cdot e^{j\frac{\pi}{2}} + i_y(t) \propto \sum_{N=1}^{\infty} 2A_N\cos(N\omega_{LO}t) + 2Q_1 A_N\cos((\omega_{RF} - N\omega_{LO})t) \quad (5)$$

It can be seen from Equation (5) that the IF signal down-converted from the image component is eliminated by coupling the two samples from the two branches with a reversed phase. Additionally, the multiple-frequency band signals can be simultaneously down-converted to the IF band below $f_{LO}/2$, which benefits from the optical sampling scheme by using an ultra-short optical pulse train. Therefore, the RF signal at the high frequency band can be down-converted to the IF band, free of image interference and without the requirement of a wide-range tunable LO signal generator. In addition, if the signal at the symmetrical frequency band with respect to $Nf_{LO}$ is desired (i.e., $\omega_{RF} < \omega_{IM}$), the interference of the image frequency component can also be suppressed by exchanging the input ports of the electrical 90° HC for quadrature coupling, i.e., the output current of the 90° HC equals $I_x(t) + I_y(t) \cdot e^{j\pi/2}$.

## 3. Simulation and Discussion

Numerical simulation is implemented to verify the feasibility of realizing broadband image-rejected down-conversion and optimize the frequency response flatness of the proposed down-conversion scheme. Here, the frequency response of the down-conversion system depends on the frequency response of the employed devices and the characteristics of the sampling optical pulse train. In the simulation, special attention is paid to the optimization of the parameter setting for the cavity-less optical pulse source. Therefore, for simplicity, the extinction ratio of the PBS, the bandwidth of the photodetectors, and the insertion loss of all devices are not taken into account in the simulation. This simplification has no influence on the evaluation of the frequency response flatness optimization. The ultra-short optical pulse source is composed of a 1550-nm DFB-LD with an output power of 16 dBm, an MZM and a PM with 3-dB bandwidths larger than 40 GHz, a spool of SMF with a GVD value of 41.4 ps/nm, and a microwave frequency synthesizer at 8 GHz. The MZM is biased at its quadrature point, and the modulation indices of the MZM and the PM are set to be $0.5\pi$ and $3\pi$, respectively. The modulation index of the PM corresponds to the GVD value of the SMF for optimal pulse compression. The responsivity of the two photodetectors is 0.6 A/W.

A single-tone microwave signal in the frequency range of 6 GHz to 39 GHz and with a power of 10 dBm is applied to the DP-DDMZM via three electrical HCs to act as the signal to be processed. After being optically sampled by the ultra-short optical pulse train, the input signal is down-converted to the IF band below 4 GHz. Figure 2a,b show the output power of the IF signals down-converted from the input signal, with its frequency varying from 6 GHz to 39 GHz. In Figure 2a,b, $I_x(t)\cdot e^{j\pi/2} + I_y(t)$ and $I_x(t) + I_y(t)\cdot e^{j\pi/2}$ are used to achieve quadrature coupling, respectively. The simulation results indicate that the power fluctuation of the IF signal is 2.69 dB within the operation frequency range of 6 GHz to 39 GHz. Meanwhile, the desired frequency bands can be switched by exchanging the IF quadrature coupling method. Figure 2c exhibits the image rejection ratios (IRRs) of the mixer with respect to the input signals at different frequencies, which are larger than 60 dB in the frequency range of 6 GHz to 35 GHz.

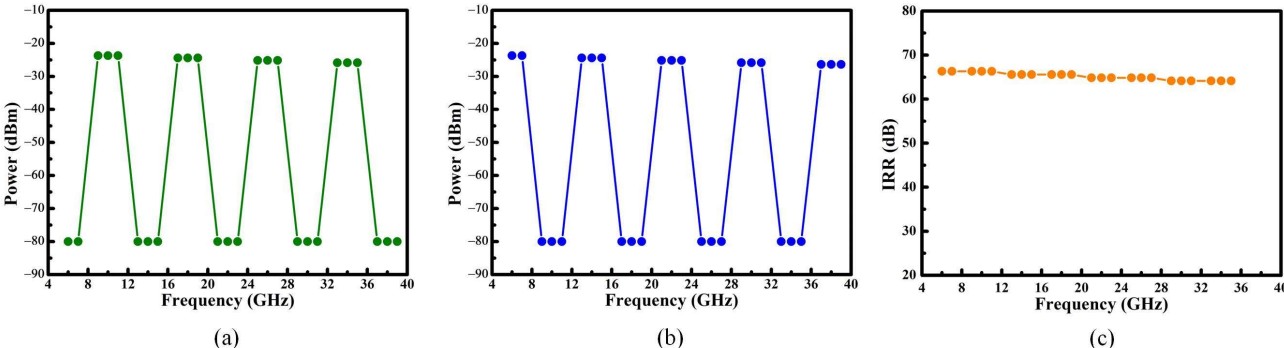

**Figure 2.** Frequency response of the image-rejected down-converter by setting the quadrature coupling method of the two photocurrents as (**a**) $I_x(t)e^{j\frac{\pi}{2}} + I_y(t)$ and (**b**) $I_x(t) + I_y(t)e^{j\frac{\pi}{2}}$. (**c**) Image rejection ratios (IRRs) of the mixer with respect to the input signals at different frequencies.

Numerical analysis is also implemented for achieving a broadband down-conversion. As shown in Equation (5), the amplitude of the IF signals is proportional to the amplitude of the frequency-multiplied component from the LO signal, with corresponding multiples. In addition, it can be seen from Equations (1)–(4) that the amplitude coefficient $A_N$ is dependent on the parameters of the ultra-short optical pulse source, including the modulation indices of the MZM and the PM, the bias phase shift of the MZM, the GVD value of the SMF, and the frequency of the LO signal. Therefore, the power fluctuation of the IF signals can be optimized through the parameters of the ultra-short optical pulse source. For an ultra-short optical pulse source with a specific repetition rate, the modulation index of the PM needs

to be set as high as possible for obtaining a broadband optical spectrum. Consequently, the GVD value of the SMF is determined accordingly. In the simulation, the power fluctuation of the IF signals down-converted from multiple frequency bands is evaluated under the condition of various modulation indices and bias phase shifts of the MZM. Figure 3a shows the power contour map of the 5th-order multiplied-frequency component at 40 GHz, where the power of the 5th-order multiplied-frequency component under the condition of $m_1 = 0.5\pi$ and $\varphi_1 = 0.5\pi$ is used as the reference value for normalization. Figure 3b shows the contour map of the power fluctuation between the 1st-order multiplied-frequency component at 8 GHz and the 5th-order multiplied-frequency component at 40 GHz. Figure 3c presents the overlapping contour map shown in Figure 3a,b. Here, the slash area corresponds to the parameter setting of the MZM to simultaneously obtain a power fluctuation below 2 dB and a normalized power of the 5th-order multiplied-frequency component at 40 GHz higher than −2 dBm. This is the optimal parameter setting for achieving broadband down-conversion.

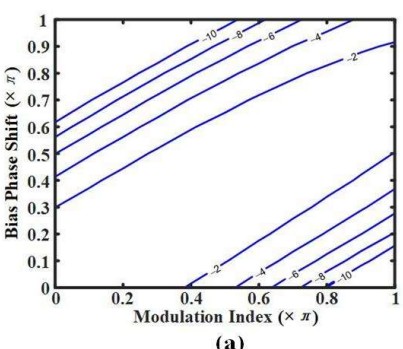 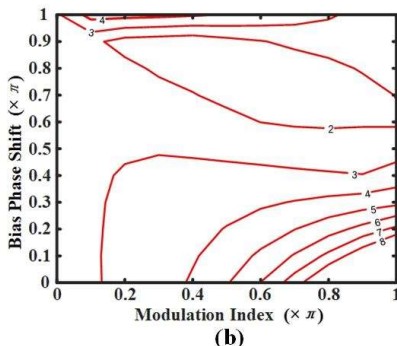 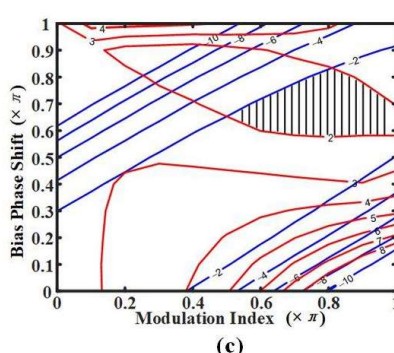

(a)    (b)    (c)

**Figure 3.** Contour maps showing (**a**) the normalized power in dB of the 5th-order multiplied-frequency component at 40 GHz, (**b**) the power fluctuation over the frequency range from 8 GHz to 40 GHz, under various modulation indices, and the bias phase shifts of the MZM. (**c**) The overlapping contour map of Figure 3a,b.

## 4. Experiment Results and Discussion

A proof-of-concept experiment is carried out to verify the feasibility of the proposed scheme. The ultra-short optical pulse source is constructed by employing a DFB-LD (Pure photonics PPCL500) at 1550 nm and with an output power of 16 dBm, a 40-Gb/s MZM (Photline, MXAN-LN-40-00-P-P-FA-FA), a 20-Gb/s PM (EOSpace, PM-SSEE-20-PFA-PFA-UV), a spool of 2.47-km-long SMF (YOFC G652D) with a GVD value of 41.4 ps/nm, and a self-developed dual-output microwave signal source with an output frequency of 8 GHz. Two pairs of electrical attenuators and RF power amplifiers are used to tune the output power of the two LO signals. Meanwhile, an EPS with a tuning range from 0° to 180° is adopted to set the phase difference of the LO signals between two outputs of the microwave signal source. In the experiment, the modulation index of the PM is set to be $3\pi$, and the modulation index of the MZM varies from 0 to $\pi$ through controlling the output power of the two LO signals, respectively. The signal profile applied to the PM is adjusted to align with each optical pulse through tuning the EPS. The input RF signal is optically sampled by the ultra-short optical pulse train via a DP-DDMZM (Fujitsu, FTM7980EDA) with a 3-dB bandwidth of 30 GHz. Here, an electrical 90° HC (Marki, QH0440) with an operation frequency range from 4 GHz to 40 GHz and two electrical 180° HCs (Gwave, GHC-180-060400) with an operation frequency range from 6 GHz to 40 GHz are used to load the RF signal onto the ultra-short optical pulse train. A PBS with an extinction ratio of 31 dB is applied to achieve polarization demultiplexing. Two photodetectors (Discovery DSH-10H) with a 3-dB bandwidth of 40 GHz and a responsivity of 0.65 A/W are employed to detect the IF signals.

To verify the optimized parameter setting for the ultra-short optical pulse source, the generated ultra-short optical pulse train is directly detected by using a broadband PD (Discovery DSH-10H), where the generated electrical spectrum is measured using an electrical spectrum analyzer (ESA, R&S FSW67). Figure 4a,b shows the normalized power of the 5th-order multiplied-frequency component at 40 GHz, and the power fluctuation between the 1st-order multiplied-frequency component at 8 GHz and the 5th-order multiplied-frequency component at 40 GHz, respectively, under different bias phase shifts of the MZM. The modulation index of the MZM is set to be $0.7\pi$. It can be seen from Figure 4 that the variation tendency of the measured power for the multiplied-frequency component at 40 GHz and the power fluctuation of the 1st-order and the 5th-order multiplied-frequency component coincide with the simulation results shown in Figure 3a,b, respectively. Specifically, the optimal parameter setting for the MZM to achieve broadband down-conversion with a higher output power is a modulation index of $0.7\pi$ and a bias phase shift of $0.7\pi$. Under the optimal parameter setting, the power flatness of the generated electrical frequency comb is 5.21 dB, which is deteriorated by approximately 3 dB compared with the simulation results due to the frequency response degradation of the PD. Figure 5 shows the optical spectrum of the generated ultra-short optical pulse train under the optimal parameter setting. The 3-dB bandwidth of the optical pulse source is measured to be 0.512 nm, indicating that the optical pulse train contains nine optical frequency comb teeth in its 3-dB bandwidth.

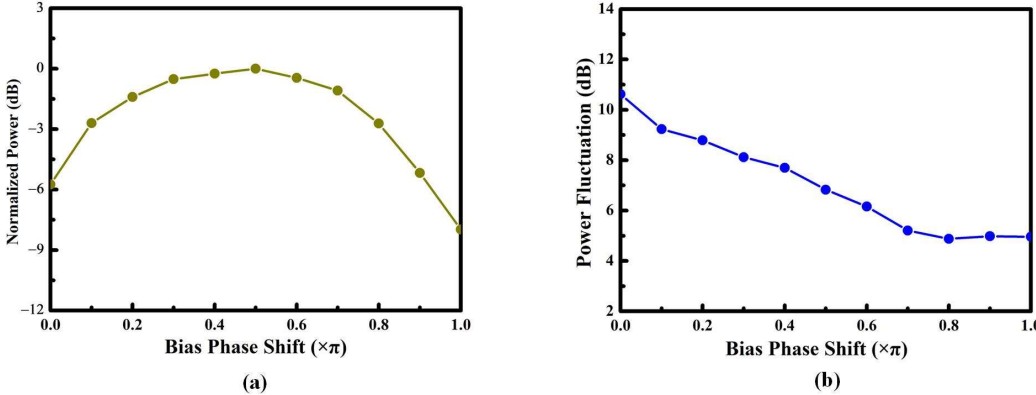

**Figure 4.** (**a**) Normalized power of the 5th-order multiplied-frequency component at 40 GHz, and (**b**) power fluctuation between the 1st-order multiplied-frequency component at 8 GHz and the 5th-order multiplied-frequency component at 40 GHz, under various bias phase shifts of the MZM.

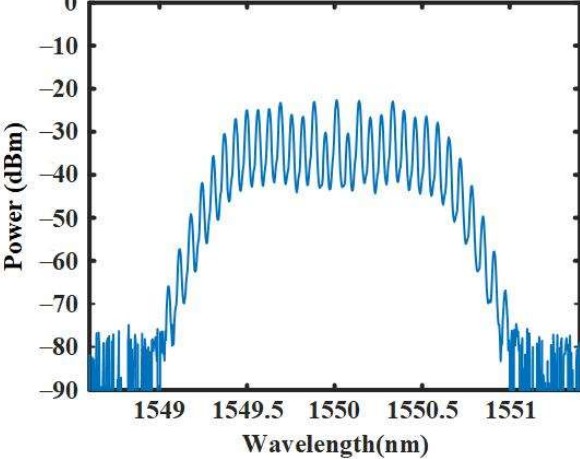

**Figure 5.** Optical spectrum of the ultra-short optical pulse train under a modulation index of $0.7\pi$ and a bias phase shift of $0.7\pi$.

Then, a single-tone RF signal in the frequency range of 6 GHz to 39 GHz and with a power of 15 dBm is generated by using a microwave frequency synthesizer (R&S SMB100A). To boost the RF signal power with a frequency coverage up to 39 GHz, two electrical amplifiers, with operation frequency ranges of 35 MHz to 22 GHz (GT-HLNA-0022G) and 22 GHz to 40 GHz (GT-LNA-2244), are adopted. After being optically sampled by the optimized ultra-short optical pulse train via the DP-DDMZM, the RF signals in the two polarization channels are detected by using two PDs, and then recorded by using a real-time oscilloscope (OSC, LeCrow 816Zi-B) with a sampling rate of 40 GS/s, a 3-dB bandwidth of 16 GHz, and a quantization bit of 8 bits. Additionally, before photodetection, two erbium-doped fiber amplifiers (EDFA, Amonics AEDFA-PA-35-B-FA) are employed to compensate for the insertion loss in the optical links. In order to correct the amplitude and phase imbalance of the signals from the two parallel branches, a compensation operation is implemented via digital signal processing (DSP), and the flow chart of the imbalance compensation is shown in Figure 6. Initially, the quantized signals from the OSC pass through two bandpass filters, respectively, to pick out the IF signals below 4 GHz. Then, the amplitude imbalance and the phase delay between the two IF signals are evaluated and compensated for in the time domain and in the frequency domain, respectively. Finally, two corrected signals are quadrature combined to obtain the IF signal, free of image interference.

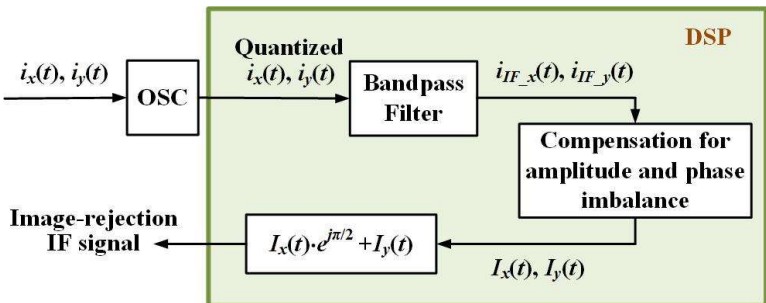

**Figure 6.** Flow chart of the imbalance compensation. OSC: oscilloscope; DSP: digital signal processing.

Figure 7a,b shows the output power of the acquired IF signal after DSP operation and quadrature coupling using $I_x(t) \cdot e^{j\pi/2} + I_y(t)$ and $I_x(t) + I_y(t) \cdot e^{j\pi/2}$, respectively, under various input RF signal frequencies. The measurement results indicate that the 6-dB mixing frequency range is from 9 GHz to 34 GHz, and from 6 GHz to 33 GHz, respectively. The power fluctuation within the frequency range of 6 GHz to 39 GHz is 8.32 dB. Compared with the power fluctuation of 2.69 dB in the simulation, the relatively large power fluctuation in the experiment is attributed to the following four factors, i.e., the electrical spectrum curve of the optical pulse source (~2 dB), the inconsistent gain of the electrical amplifier (~2 dB), and the frequency response degradation of the DP-DDMZM (~3 dB) and that of the three HCs (~3 dB). Using to the measurement results shown in Figure 7, the IRRs of the mixer with input signals at different frequencies are calculated and presented in Figure 8. The IRR of the mixer is measured to be larger than 48.91 dB, indicating that the proposed scheme can achieve effective image rejection. In this condition, the IRR of the mixer is restrained by the noise floor of the electrical spectrum, where the noise floor level depends on the acquisition time of 2 μs limited by the storage depth of the OSC (corresponding to a spectral resolution of 500 kHz). In addition, to further improve the IRR, the input optical power of the two PDs should be increased. In our experiment, the input optical power is restrained below −2 dBm due to the damage threshold of the PDs.

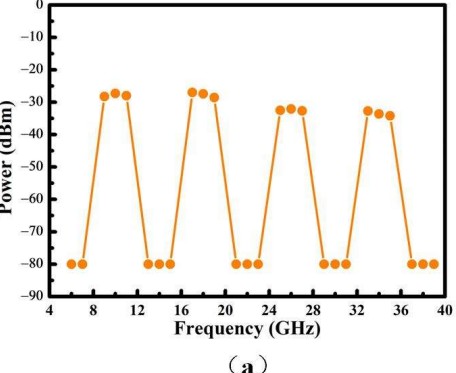 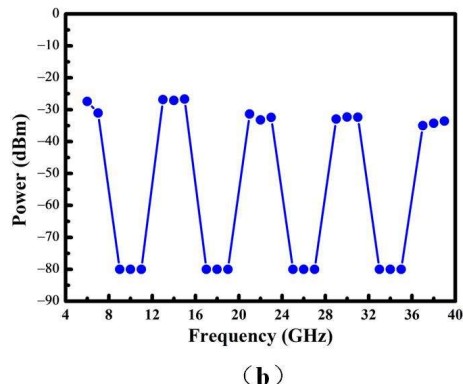

(a)　　　　　　　　　　　　　　　　(b)

**Figure 7.** Output power of the IF signal after the RF signal is down-converted and with the DSP operation, where the quadrature coupling method of the two polarized photocurrents are set to be (**a**) $I_x(t)\cdot e^{j\pi/2} + I_y(t)$ and (**b**) $I_x(t) + I_y(t)\cdot e^{j\pi/2}$, respectively.

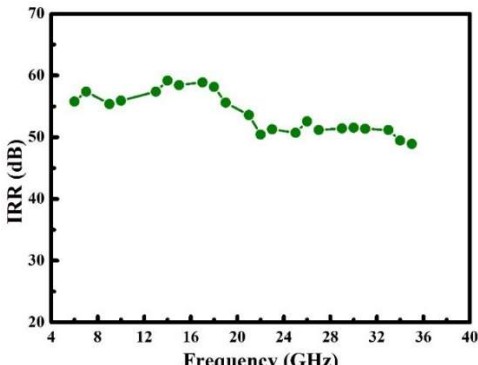

**Figure 8.** Image rejection ratios (IRRs) of the mixer under various input RF signals at different frequencies.

To verify the capacity of real-time image rejection, the correction operation for the amplitude and phase imbalance between the two parallel branches are also implemented in the analog domain using an electrical power attenuator (Rebes, RBS-11-4.3-2) with an operation frequency range from DC to 4.3 GHz and an electrical phase shifter (Connphy, MPS-DC18G-60S) with an operation frequency range from DC to 18 GHz. After compensation, an electrical 90° HC (Marki, QH-0R714), with an operation frequency range from 0.7 GHz to 14.5 GHz, is used to quadrature combine the two parallel signals. The output IF signal is characterized by using an ESA. In this section, the RF signals and their corresponding image signals at four diverse frequency bands and with a power of 11 dBm are generated by the microwave frequency synthesizer and then down-converted through the proposed mixer. Figure 9a–d shows the electrical spectra of the IF signals down-converted from the RF signals at 10 GHz, 18 GHz, 22 GHz, and 30 GHz, along with those down-converted from their corresponding image signals at 6 GHz, 14 GHz, 26 GHz, and 34 GHz, respectively, where the resolution bandwidth of the ESA is set to be 100 kHz. The IRRs of the signals located at the four diverse frequency bands are 38.89 dB, 44.01 dB, 41.21 dB and 50.13 dB, respectively, indicating that the system is capable of achieving image rejection for multi-band frequency down-conversion. Compared with the results through using DSP, the measured IRRs value after compensation in the analog domain are deteriorated, which is mainly attributed to the residual amplitude and phase deviation between the two parallel branches. Therefore, the residual amplitude deviation is derived from the 0.1 dB tuning step of the electrical power attenuator and the ±0.8 dB amplitude imbalance of the adopted 90° HC. The phase deviation originates from the ±8° phase offset of the adopted 90° HC. In addition, the electrical amplifier used for boosting the input signal is removed, leading

to the power reduction of the measured IF signal, which also results in the deterioration of the IRR value.

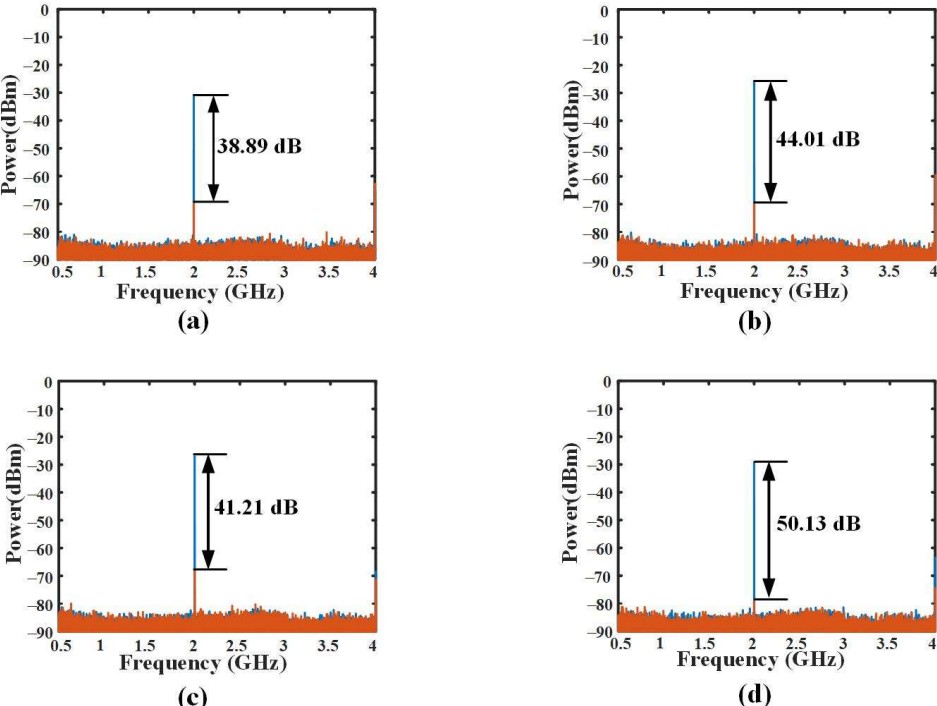

**Figure 9.** Electrical spectra of the IF signals down-converted from (**a**) the RF signal at 10 GHz and the corresponding image signal at 6 GHz, (**b**) the RF signal at 18 GHz and the corresponding image signal at 14 GHz, (**c**) the RF signal at 22 GHz and the corresponding image signal at 26 GHz, (**d**) the RF signal at 30 GHz and the corresponding image signal at 34 GHz. (RF signal: blue line; image signal: red line).

It should be noted that the instantaneous bandwidth of the constructed multi-band down-converter is limited to 4 GHz, since the photonic sampling rate is 8 GS/s. Nevertheless, this instantaneous bandwidth is large enough for most of the current application scenarios in wireless communications and radar systems because the signal bandwidth for a single frequency band in these applications is generally in the range of tens of MHz to hundreds of MHz. To accommodate large bandwidth applications, the sampling rate can be easily increased in the proposed scheme by using an LO signal with a higher frequency and simultaneously varying the GVD value of the dispersion medium to guarantee chirp compensation in the cavity-less ultra-short optical pulse source. In addition, the multi-band RF signals are all down-converted to the Nyquist bandwidth below $f_{LO}/2$. Hence, the signals after down-conversion may be confronted with spectrum overlapping. This can be avoided by carefully designing the carrier frequencies in the RF transmitter, or by properly tuning the sampling rate of the down-converter.

Finally, it should be pointed out that the LO signal employed to drive the cavity-less optical pulse source is generated by using a self-developed microwave signal source. The phase noise of the LO signal is inferior to that of an optoelectronic oscillator (OEO) [32–34]. Therefore, to achieve a high-performance broadband down-conversion without degrading the phase noise performance of the multi-band RF signals, OEOs are promising candidates to generate LO signals for driving the cavity-less optical pulse source in the proposed down-conversion scheme.

## 5. Conclusions

In summary, we have experimentally demonstrated a multi-band down-conversion scheme with image rejection over a large frequency range. In the proposed scheme, the RF signals at various frequency bands can be simultaneously down-converted to the IF band after being optically sampled by an ultra-short optical pulse train via a DP-DDMZM and detection using a pair of low-speed PDs. Meanwhile, the image component can be eliminated by the photonic-assisted Hartley method. In the numerical simulation, the optimal parameter setting of the MZM in the ultra-short optical pulse source with a repetition rate of 8 GHz was obtained to achieve a broadband down-conversion. In the experiment, based on the optimized optical pulse source, multi-band RF signals over the frequency range of 6 GHz to 39 GHz were down-converted to the IF band below 4 GHz by employing a DP-DDMZM with a 3-dB bandwidth of 30 GHz, where the power flatness was measured to be 8.32 dB. With the assistance of DSP compensation for the amplitude and phase imbalance of the parallel branches, the image components were suppressed, and the IRR was measured to be larger than 48.91 dB. In addition, the capacity of real-time image rejection was also verified by adopting a 90° HC to compensate for the imbalance of the parallel paths in the analog domain. Since a pre-selection filter for use before down-conversion dose not currently exist, the proposed image-rejected multi-band frequency down-conversion scheme can be used in broadband wireless communication and radar systems.

**Author Contributions:** Conceptualization, L.X. and D.P.; methodology, D.P. and Y.Q.; software, L.X.; validation, L.X. and D.P.; formal analysis, J.L. and M.X.; investigation, L.X., D.P. and S.F.; resources, Y.Q.; data curation, D.P. and O.X.; writing—original draft preparation, L.X.; writing—review and editing, D.P., Y.Q. and S.F.; supervision, J.L.; project administration, O.X.; funding acquisition, Y.Q. and D.P. All authors have read and agreed to the published version of the manuscript.

**Funding:** Funding was provided by the National Key Research and Development Program of China (2018YFB1801001), the National Natural Science Foundation of China (62175038), the Open Fund of the Guangdong Provincial Key Laboratory of Optical Fiber Sensing and Communications (Jinan University), and the Guangdong Introducing Innovative and Entrepreneurial Teams of "The Pearl River Talent Recruitment Program" (2019ZT08X340).

**Institutional Review Board Statement:** Not applicable.

**Informed Consent Statement:** Not applicable.

**Data Availability Statement:** Data underlying the results presented in this paper are not publicly available at this time, but may be obtained from the authors upon reasonable request.

**Conflicts of Interest:** The authors declare no conflict of interest.

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
