# Peer review of "Image-Rejected Multi-Band Frequency Down-Conversion Based on Photonic Sampling"

_photonics, doi:10.3390/photonics10010035_

Round 1

Reviewer 1 Report

The authors have proposed and experimentally demonstrated a broadband image-rejected microwave frequency down-converter based on Hartley architecture. The main innovation of this scheme lies in the use of photonic sampling to realize broadband down-conversion without the requirement of a wide-range tunable local oscillation (LO) signal generator. Both simulation and experiment have been carried out to demonstrate the feasibility of the proposed scheme. Especially, the image rejection performance has been demonstrated in both the digital domain and the analog domain. The presented results are convincing, and the manuscript is well-organized. Hence, it can be accepted for publication in Photonics if the authors consider the following comments.

(1)    In the experiment, the frequency of the LO signal is 8 GHz, which limits the instantaneous bandwidth to 4 GHz. Is this value large enough for the multi-band frequency down-conversion application?

(2)    If multi-band radio-frequency (RF) signals are applied to the down-converter, they are all down-converted to the Nyquist bandwidth below 4 GHz. How to avoid spectrum overlapping after down-conversion?

Reviewer 2 Report

The authors propose and demonstrate a broadband microwave photonic frequency down-conversion scheme based on photonic sampling, which is featured with multi-band operation and image rejection. In this manuscript, the background of multi-band frequency down-conversion and the recent advance of microwave photonic frequency down-converters are adequately introduced and analyzed. The operation principle of the proposed scheme is presented in detail. Numerical simulation is implemented to obtain the optimal parameter setting of the cavity-less ultra-short optical pulse source. Based on the optimized parameters, multi-band frequency down-conversion is experimentally demonstrated in the frequency range of 6 GHz to 39 GHz, where the image rejection ratios are beyond 48 dB. In summary, this is a perfect work, and the results are convincing. Therefore, I believe that this manuscript is recommended for publication in its present form.

Reviewer 3 Report

The Authors propose new method to perform the frequency down-conversion without using directly a filter and delating the image signals.  Although the results are promising for the next generation of communication systems, a major revision of the manuscript is suggested to address the following comments:

-          In the caption of Fig. 1, there are not references to the single images (a-f). It is recommended to add them to make the figure easy to be read and understood.

-          The Authors should clarify how the LO signal is generated. The LO signal seems to be generated by using external circuit. This solution should be justified with respect OEO solutions that guarantee low phase noise values (see, Chip-Scaled Ka-Band Photonic Linearly Chirped Microwave Waveform Generator. Frontiers in Physics, 158, (2022); Arbitrary Microwave Waveform Generation Based on a Tunable Optoelectronic Oscillator. J Lightwave Technol (2013);  Effect of Laser Coupling and Active Stabilization on the Phase Noise Performance of Optoelectronic Microwave Oscillators Based on Whispering-Gallery-Mode Resonators. IEEE Photon J 7(1):1–11, 2014).

-          In the Simulation section, the losses has completely neglected. There are a lot lossy elements in the proposed architecture as the Beam Splitter, the polarization rotator, the phase shifter that could affect the signal to noise ratio.

-          In the Simulation section, the noise figure has not considered. The electrical amplifier enhances also the noise level and it deserves attention. Its position in the structure has to be justified through simulations. 

-          In the Experiment section, the Authors state that, due to losses, the system needs to be compensated through EDFAs and DSP. In particular, in the DSP element there is a filter so that it is not a filter-free solution actually.

-          In the Experiment section, at lines 335-336, the Authors refer to four factor that make different the simulations to the experimental results but you do not take in account the loss contributes too.

Round 2

Reviewer 3 Report

The Authors have modified the manuscript according to the Reviewer suggestions.